# Analysis of SiC/Si Heterojunction Band Energy and Interface State Characteristics for SiC/Si VDMOS

**DOI:** 10.3390/mi14101890

**Published:** 2023-09-30

**Authors:** Xin Yang, Baoxing Duan, Yintang Yang

**Affiliations:** Key Laboratory of the Ministry of Education for Wide Band-Gap Semiconductor Materials and Devices, School of Microelectronics, Xidian University, No. 2 South TaiBai Road, Xi’an 710071, China; bxduan@163.com (B.D.); ytyang@163.com (Y.Y.)

**Keywords:** heterojunction, MOSFETs, electric field, energy band, parameter, capacitance

## Abstract

SiC/Si and GaN/Si heterojunction technology has been widely used in power semiconductor devices, and SiC/Si VDMOS and GaN/Si VDMOS were proposed in our previous paper. Based on existing research, breakdown point transfer technology (BPT) was used to optimize SiC/Si VDMOS. Simulation results showed that the *BV* of the SiC/Si heterojunction VDMOS was considerably increased from 259 V to 1144 V, and *R_on,sp_* decreased from 18.2 mΩ·cm^2^ to 6.03 mΩ·cm^2^ compared with Si VDMOS. In order to analyze the characteristics of the SiC/Si heterojunction structure deeply, the influence of the interface state characteristics of the SiC/Si heterojunction on the electrical parameters of VDMOS was analyzed, including electric field characteristics, blocking characteristics, output characteristics, and transfer characteristics. In addition, the influence of the interface state of the SiC/Si heterojunction on energy band characteristics was analyzed. The results showed that with an increase in the interfacial charge (acceptor) concentration, the p-type trap layer was introduced into the interface of the SiC/Si heterojunction, energy increased slightly, and the barrier height difference at the heterojunction increased, resulting in an increase in *BV*. At the same time, since the barrier height became higher, electrons did not flow easily, so *R_on,sp_* increased. On the contrary, when a charge (donor) was introduced at the interface of the SiC/Si heterojunction, the number of electrons in the channel increased, resulting in an increase in the electron current, which is conducive to the flow of electrons, resulting in a decrease in *R_on,sp_*. The energy band and other characteristics of devices with temperature were simulated at different temperatures. Finally, the effects of SiC/Si heterojunction interface states on interface capacitances and switching performances of VDMOS devices were also discussed.

## 1. Introduction

A vertical double-diffusion metal oxide semiconductor (VDMOS) is an important component in the field of power semiconductor devices; due to its fast switching speed, low loss, high input impedance, low driving power, and excellent frequency characteristics, it has been widely used in power integrated circuits and power integrated systems [1,2,3,4,5,6,7,8,9,10]. However, the main problem of VDMOS power devices is that the specific on-resistance (*R_on,sp_*) of the device increases sharply with an increase in breakdown voltage (*BV*), which greatly limits the development and application of VDMOS power devices [11,12,13,14,15,16,17,18,19,20,21,22,23,24,25,26,27,28]. 

However, efforts to improve Si power devices are always limited by the low critical electric field of Si, which gives an opportunity for the development of SiC devices. SiC materials have a large bandgap and a critical electric field about 10 times that of Si, which can break the limit of Si materials [25,26,27,28,29,30,31,32,33,34,35,36,37,38,39,40,41,42,43,44,45,46,47]. Nevertheless, SiC power devices suffer from gate oxide reliability and some difficulties in manufacturing processes, such as the diffusion of impurity and the realization of high-quality ohm contact [16,17]. Moreover, manufacturing SiC devices is much more costly compared with Si devices. The successful fabrication of SiC/Si substrates offers a practical approach to solving these problems [18,19,20,21,22]. 

A SiC/Si heterojunction VDMOS combines the mature process of Si materials with the wide bandgap of SiC materials. Thus, a high critical breakdown electric field can optimize *BV* compared with a traditional Si-based power device [16,17]. The contradiction between *BV* and *R_on,sp_* is optimized, and the high thermal conductivity of SiC materials is beneficial to the heat dissipation of a VDMOS device, which effectively improves the performance of the device. Since the active region of the device is formed from silicon semiconductor material, a mature silicon process can be employed in the device fabrication process to achieve better ohmic contact [16,17,18,19,20,21,22,23,24,25]. SiC/Si VDMOS and GaN/Si VDMOS were proposed in our previous paper, and the influence of device electrical and structural parameters was studied [16,18]. However, this paper mainly discusses the influence of the interface state on the electric field, band energy distribution, temperature, and switching characteristics.

## 2. Materials and Methods

In this paper, a VDMOS with SiC/Si heterojunction was used to optimize *BV* by breakdown point transfer (BPT), which transfers the breakdown point from a high electric field to a low electric field. Figure 1 shows a cell of the proposed SiC/Si heterojunction VDMOS; the formation of the Si/SiC substrate can be realized by the method in [22]. “*D_Si_*” is defined as the Si thickness, “*L_D_*” is the length of N^−^ drift region, and “*N_D_*” is the concentration of N^−^ drift region in the structures.

In this paper, a two-dimensional numerical simulation of SiC/Si VDMOS is performed using ISE-TCAD. The main physics models were applied in Synopsys SentaurusTM tools simulation, including Mobility (DopingDep High Field Sat Enormal), EffectiveIntrinsic Density (OldSlotboom), and Recombination (SRH (DopingDep) and Auger Avalanche (Eparal)). The criterion of breakdown was BreakCriteria {Current (Contact = “drain” Absval = 1 × 10^−7^)}. The main solving model was Coupled {Poisson Electron Hole}. For the coordinates, it was necessary to optimize the parameters in the numerical simulations. Some of the device parameters in the simulation are presented in Table 1. The ambient temperature was 300 K, the breakdown voltage (*BV*) was obtained at *V_GS_* = 0 V, and the specific on-resistance (*R_on,sp_*) was obtained at *V_GS_* = 10 V; the simulation results of the four devices are shown in Table 2.

## 3. Results and Discussion

### 3.1. The Influence of Interface State on Electrical Parameters

The vertical electric field for SiC/Si VDMOS, Si VDMOS, and SiC VDMOS are shown in Figure 2a. For SiC VDMOS, *BV* reached 1747V. For Si VDMOS, when X = 2.5 μm (shown in Figure 1), the maximum field strength was 3.03 × 10^5^ V/cm (reaching the critical breakdown electric field of Si materials), and *BV* was 259 V of Si VDMOS. For SiC/Si VDMOS, when the drain voltage reached 259 V, the electric field strength at the interface between the P-base and the N-type drift regions did not reach the critical breakdown field strength of SiC, so the device did not break down. As the drain voltage was further increased, the electric field strength of the device increased until the electric field at the heterojunction reached 3.45 × 10^5^ V/cm (reaching the critical breakdown field of the Si materials), and the SiC/Si VDMOS broke down. Therefore, the *BV* of SiC/Si VDMOS was increased from 259 V to 1144 V compared with the conventional Si VDMOS [16,17]. According to previous experimental results [23,24], SiC/Si interfacial charges were introduced during the direct bonding process. The effect of different interface state charge concentrations on the vertical electric field of SiC/Si VDMOS is shown in Figure 2b. The type of interface state charge introduced was donor (electron) or acceptor (hole) in this paper. An increase in the interface state charge (acceptorlike) concentrations resulted in an increase in the vertical electric field of the SiC/Si VDMOS compared with SiC/Si VDMOS without an interface state charge. This is because of the p-type trap layer introduced by the interface charge (acceptor) at the SiC/Si heterojunction, which enhanced the internal electric field at the SiC/Si interface and changed the distribution of the electric field. As the interface state charge (donor) concentration increased, the vertical electric field at the SiC/Si heterojunction decreased.

The optimized blocking characteristics and output characteristics for the SiC/Si VDMOS are provided in Figure 3a,b. As can be seen, *BV* was 1144 V, and *R_on,sp_* was 6.03 mΩ·cm^2^ for the SiC/Si VDMOS without an interface state charge. The *BV* of the SiC/Si VDMOS increased from 1182 V to 1507 V with increasing interface state charges (acceptorlike) concentrations. This is because the vertical electric field of SiC/Si VDMOS increased by SiC/Si heterojunction, which the p-type trap layer introduced by the interface charge (acceptor) at the SiC/Si heterojunction, resulting in *R_on,sp_* increasing to 28 mΩ·cm^2^. However, the internal electron barrier was induced in the inversion layer at the SiC/Si interface, and the number of electrons in the channel increased, resulting in an increase in the electron current. This resulted in *R_on,sp_* dropping from 6.03 mΩ·cm^2^ to 5.80 mΩ·cm^2^, and *BV* decreased to 748 V at this time. The transfer characteristics for the SiC/Si VDMOS are shown in Figure 3c. It is reported that the charge at the SiC/Si interface had an influence on the I–V characteristics of the SiC/Si VDMOS. That is, when the interface charge donor (electron) concentration increased, the threshold voltage (*V_TH_*) of SiC/Si VDMOS increased from 4.82 V to 5.08 V. When the interface charge was considered an acceptor (hole), the *V_TH_* of the SiC/Si VDMOS decreased from 4.75 V to 4.51 V as the interface charge concentration increased from 1 × 10^5^/cm^2^ to 5 × 10^5^/cm^2^.

### 3.2. The Influence of Interface State on Band Energy Distributions

Figure 4 shows the energy bands of SiC/Si VDMOS, Si VDMOS, and SiC VDMOS during thermal equilibrium. It can be seen that for Si VDMOS, the bandgap was 1.12 eV, and the curve X = 2.5 μm was the focus of the Si VDMOS breakdown voltage (shown in Figure 1), which means that the breakdown point was located at the junction of the P-based and N-type drift regions, the site of the maximum radius of curvature of conventional Si VDMOS. Since the band gap of SiC VDMOS was 3.26 eV, curve X = 4.9 μm was the focal point of the SiC VDMOS breakdown voltage, and the *BV* of the device reached 1747 V. The curve X = 4.9 μm was the focus of the breakdown voltage for SiC/Si VDMOS with SiC/Si heterojunction (shown in Figure 1). Due to the high critical breakdown field strength of SiC materials, *BV* increased from 259 V to 1144 V compared with Si VDMOS.

Figure 5 shows the band diagram of N^−^–Si/N–SiC in the middle of mesa (AA’), the P–Si/N–SiC heterojunction (BB’), the P–Si/P–SiC heterojunction (CC’), and the N^+^–Si/P–SiC heterojunction (DD’). To analyze the band state of the SiC/Si VDMOS heterojunction, four tangents in the y direction (AA’, BB’, CC’, and DD’) were chosen. We can see that the SiC/Si VDMOS heterojunction is affected not only by the two materials Si and SiC with different bandgaps but also by the doping concentration and different doping types at different regions of the device, which change the heterojunction bandgap of SiC/Si VDMOS.

The energy band diagram of SiC/Si VDMOS with different interface state charges (acceptorlike) is shown in Figure 6a. In this figure, as the interface concentration increases, the energy at the SiC/Si heterojunction rises slightly due to the p-type trap layer introduced by the interface charge (acceptor) at the SiC/Si heterojunction interface. This results in an increase in the barrier height difference at the heterojunction, resulting in an increase in the *BV* of SiC/Si VDMOS (shown in Figure 3a). Further, since the height of the barrier becomes high, electrons do not easily flow, and thus, *R_on,sp_* increases (shown in Figure 3b). Figure 6b shows the influence of different interface state charges (donors) on the energy band diagram of SiC/Si VDMOS. Electrons were induced in the inversion layer at the SiC/Si heterojunction interface. As the interface concentration increased, the number of electrons in the channel increased, resulting in an increase in the electron current. The decrease in the barrier height resulted in a decrease in the *BV* of SiC/Si VDMOS, which facilitated electron flow and caused a decrease in *R_on,sp_* (shown in Figure 3a,b). 

### 3.3. The Influence of Interface State on Temperature Effects

Figure 7a shows the vertical electric field distributions for SiC/Si VDMOS at the temperature range of 300 K to 360 K. It can be seen that the highest electric field of the SiC/Si VDMOS reached 9.8 MV/cm, and *BV* was 1144 V at 300 K [25,26,27,28,29]. When the temperature was increased to 360 K, the maximum electric field of the SiC/Si VDMOS dropped to 8.8 MV/cm, and *BV* dropped to 1000 V. The dependences of *BV*, *R_on,sp_*, and figure-of-merit (FOM = *BV*^2^/*R_on,sp_*) changes from 300 K to 360 K for the SiC/Si VDMOS are shown in Figure 7b. It was found that the *BV* of SiC/Si VDMOS decreased from 1144 V to 1000 V as the temperature increased. In addition, the *R_on,sp_* of SiC/Si VDMOS increased, yielding a FOM (217 MW/cm^2^) of SiC/Si VDMOS that dropped to 48 MW/cm^2^. 

Figure 8a shows the band diagram of the P-Si/N−SiC heterojunction at a temperature range of 300 K to 360 K. The barrier height difference decreased when the temperature rose and *BV* decreased. At the same time, the resistivity and impurity ionization rate of the drift region increased as the temperature increased [37]. Transfer characteristics changes from 300 K to 360 K for the SiC/Si VDMOS are shown in Figure 8b. At 300 K, *V_TH_* was 4.37 V for SiC/Si VDMOS. As the temperature increased, the barrier height difference decreased, resulting in a decrease in the *V_TH_* of SiC/Si VDMOS. When the temperature reached 360 K, the *V_TH_* of the SiC/Si VDMOS was reduced to 3.68 V.

### 3.4. The Influence of Interface State on Interface Capacitances and Switching Performances

Figure 9 shows the reverse transfer capacitances (*C_rss_*) of SiC/Si VDMOS, Si VDMOS, and SiC VDMOS. *C_rss_* is especially important when VDMOS devices are used for power supplies due to the switching loss of the device being seriously affected. The expression for *C_rss_* is:(1)$$C_{rss}=C_{gd}$$

As shown in Figure 9, the *C_rss_* of SiC/Si VDMOS was smaller than that of conventional Si VDMOS and SiC VDMOS. A new capacitor was introduced at the interface where the SiC/Si heterojunction was introduced, thereby reducing the depletion capacitance of the SiC/Si VDMOS.

Figure 10a shows the influence of different interface state charges (acceptorlike) on the reverse transfer capacitances (*C_rss_*) for SiC/Si VDMOS. When the acceptor (hole) interface charge was considered, an opposite type of negative charge was induced at the channel of e SiC/Si VDMOS. This increased the depletion charge of the device, causing the *C_rss_* of SiC/Si VDMOSFET to increase as the interface concentration increased. The *C_rss_* of SiC/Si VDMOS with different interface state charges (donor) is shown in Figure 10b. A positive charge induced at the channel of the SiC/Si VDMOS reduced the depletion charge of SiC/Si VDMOS due to the interface charge donor (electron) being introduced at the SiC/Si interface. Therefore, as the concentration increased, the *C_rss_* of SiC/Si VDMOS decreased.

Figure 11a shows the dynamic performance of the three devices under resistive load. It can be seen that the turn-OFF speed of SiC VDMOS was lower than that of SiC/Si VDMOS and Si VDMOS. The dynamic characteristics of the proposed device were more suitable for low and medium frequencies in the power field, while the turn-ON speed for the two devices was almost the same. Figure 11b shows the influence of different interface state charges on the switching characteristics for SiC/Si VDMOS. A positive charge was induced at the channel of the SiC/Si VDMOS due to the interface charge donor (electron) being introduced. This reduced the depletion charge of the device, resulting in a faster turn-OFF speed of SiC/Si VDMOS. When the acceptor (hole) interface charge was considered, the depletion charge of the device increased. This resulted in a slower turn-OFF of SiC/Si VDMOS, while the turn-ON speed for the three cases was almost the same. The dynamic characteristics of the SiC/Si VDMOS were more suitable for low and medium frequencies in the power field.

The key processes for the feasibility of manufacturing SiC/Si VDMOS are shown in Figure 12, and some key processes are similar to Si VDMOS. The formation of the SiC/Si drift region can be achieved by the method in Ref. [22]. The simplified wafer bonding process is as follows: (a) ion implantation forms a P-well in the N–SiC drift region; (b) ion implantation to form source region; (c) epitaxial growth of N–Si Layer; (d) wafer bonding is carried out and ion implantation is made to form the portion of the P-well in the N–Si layer; (e) ion implantation to form the portion of the source in the N–Si layer; and (f) the gate oxide layer grows, the field oxide layer grows, and then the source electrode and the drain electrode are contacted.

## 4. Conclusions

SiC/Si VDMOS and GaN/Si VDMOS were proposed in our previous paper; however, only the electrical and structural parameters of the device were studied. In the present study, SiC/Si VDMOS was optimized based on breakdown point transfer technology, and the influence of the band interface state on the device was analyzed. The *BV* of the SiC/Si heterojunction VDMOS was considerably increased from 259 V to 1144 V, and *R_on,sp_* decreased from 18.2 mΩ·cm^2^ to 6.03 mΩ·cm^2^. Simulation results showed that the type and concentration of interface charges have a great influence on the electric field, blocking characteristics, output characteristics, transfer characteristics, and energy band distributions of SiC/Si VDMOS but have little effect on the capacitance characteristics and switching performances. As the positive concentration acceptor (hole) increased, the energy at the SiC/Si heterojunction rose slightly, resulting in an increase in *BV*. As the negative concentration donor (electron) increased, a decrease in the barrier height resulted in a decrease in the *R_on,sp_* of *BV*. The electrical characteristics of SiC/Si VDMOS were changed when the temperature was raised, resulting in a negative temperature coefficient for *BV* and *V_TH_* but a positive temperature coefficient for *R_on,sp_*. 

## Figures and Tables

**Figure 1 micromachines-14-01890-f001:**
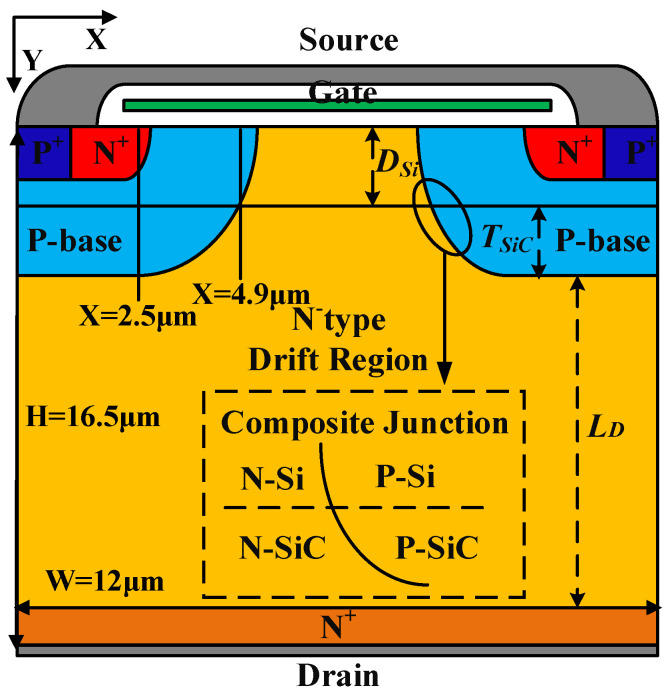
Cross-section of the novel SiC/Si VDMOS.

**Figure 2 micromachines-14-01890-f002:**
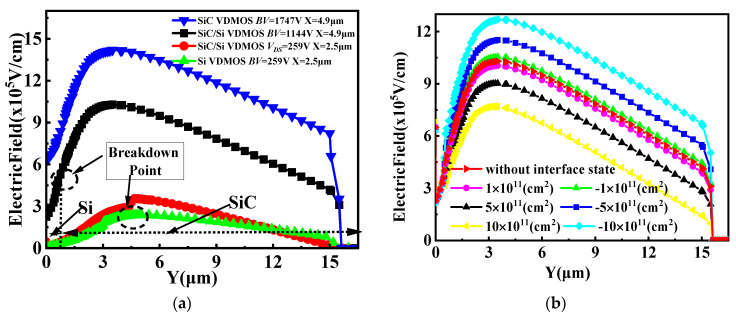
(**a**) Vertical electric field distributions for SiC/Si VDMOS, Si VDMOS, and SiC VDMOS, and (**b**) the influence of different interface state charges (acceptorlike and donor) on the vertical electric field of SiC/Si VDMOS: *L_D_* = 5 μm, *D_Si_* = 0.5 μm.

**Figure 3 micromachines-14-01890-f003:**
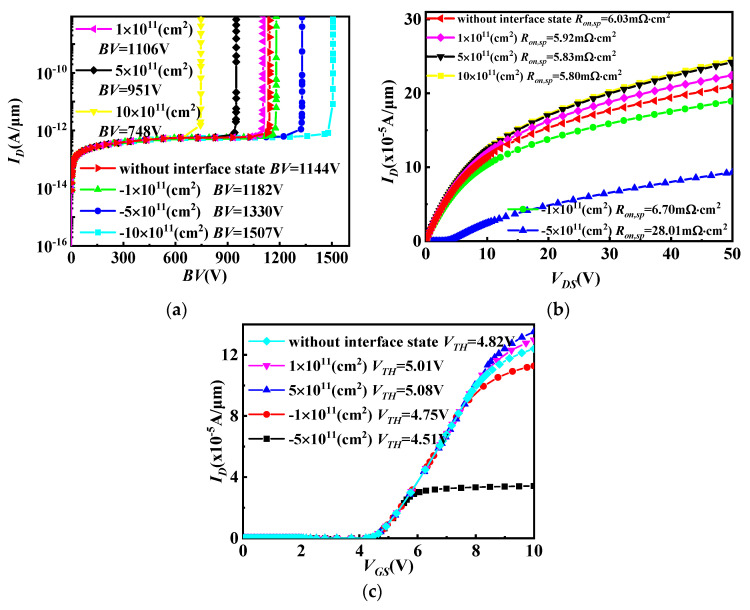
(**a**) Blocking characteristics, (**b**) output characteristics, and (**c**) transfer characteristics for SiC/Si VDMOS with the different concentrations of interface state charge.

**Figure 4 micromachines-14-01890-f004:**
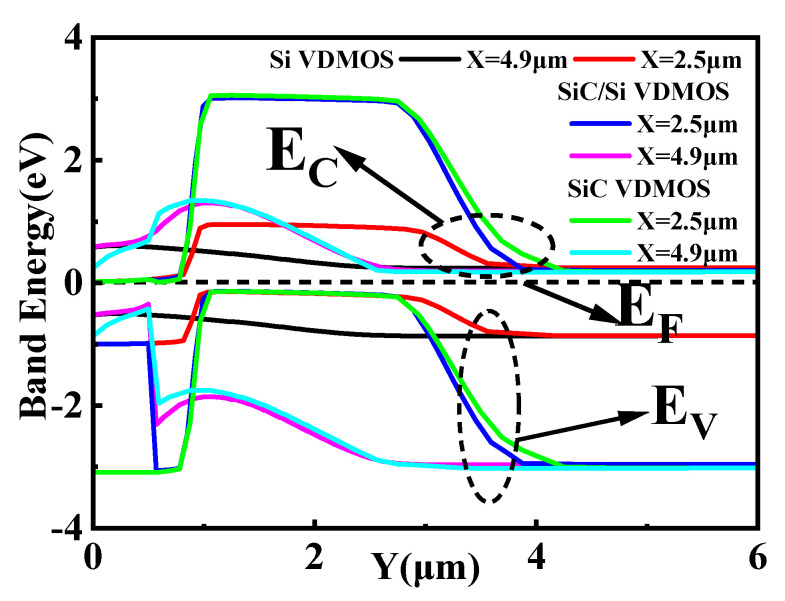
The energy band diagram at thermal equilibrium for SiC/Si VDMOS, Si VDMOS, and SiC VDMOS.

**Figure 5 micromachines-14-01890-f005:**
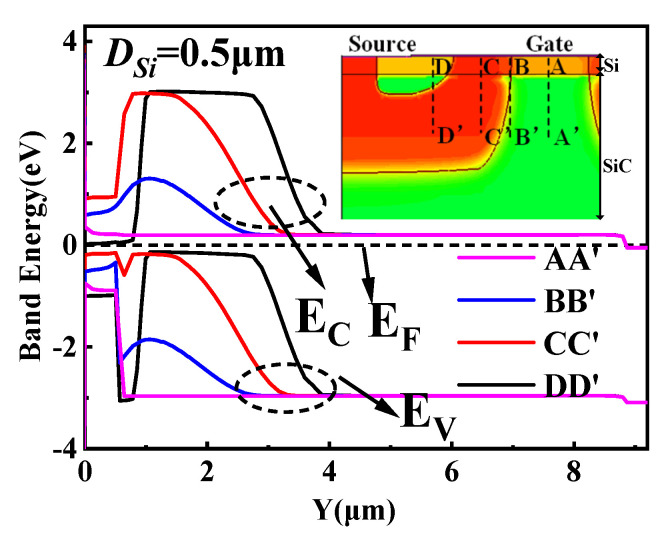
Band diagram of N^−^-Si/N^−^-SiC in the middle of mesa (AA’), the P-Si/N^−^-SiC heterojunction (BB’), the P-Si/P-SiC heterojunction (CC’), and the N^+^-Si/P-SiC heterojunction (DD’).

**Figure 6 micromachines-14-01890-f006:**
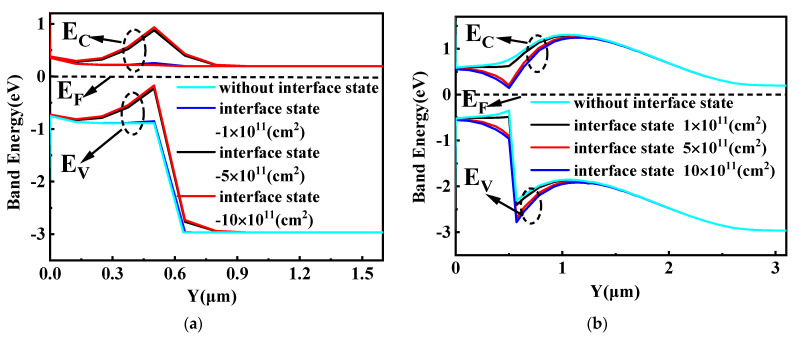
(**a**) The influence of different interface state charges (acceptorlike) on the energy band diagram of SiC/Si VDMOS, and (**b**) the influence of different interface state charges (donor) on the energy band diagram of SiC/Si VDMOS: *L_D_* = 5 μm, *D_Si_* = 0.5 μm.

**Figure 7 micromachines-14-01890-f007:**
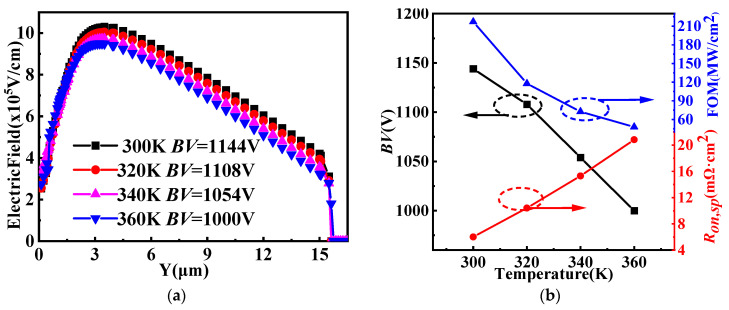
(**a**) Vertical electric field distributions for SiC/Si VDMOS at a temperature range of 300 to 360 K, and (**b**) dependences of *BV*, *R_on,sp_*, and figure-of-merit (FOM = *BV*^2^/*R_on,sp_*) changes from 300 to 360 K for SiC/Si VDMOS.

**Figure 8 micromachines-14-01890-f008:**
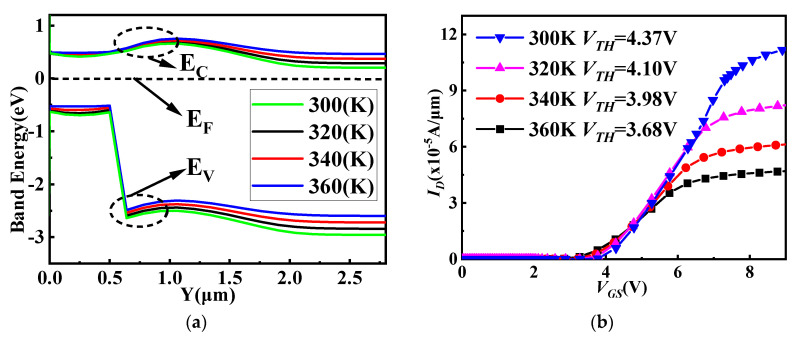
(**a**) Band diagram of the P-Si/N^−^-SiC heterojunction at a temperature range of 300 to 360 K, and (**b**) transfer characteristic changes from 300 K to 360 K for SiC/Si VDMOS.

**Figure 9 micromachines-14-01890-f009:**
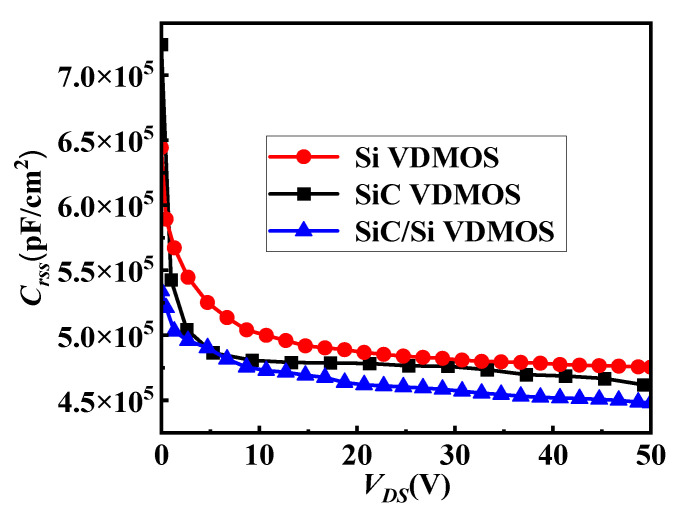
Reverse transfer capacitances (*C_rss_*) of SiC/Si VDMOS, Si VDMOS, and SiC VDMOS.

**Figure 10 micromachines-14-01890-f010:**
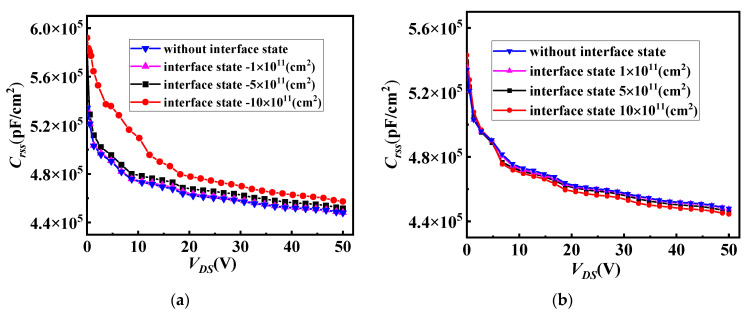
(**a**) The influence of different interface state charges (acceptorlike) on the reverse transfer capacitances (*C_rss_*) for SiC/Si VDMOS, and (**b**) the influence of different interface state charges (donor) on the reverse transfer capacitances (*C_rss_*) for SiC/Si VDMOS.

**Figure 11 micromachines-14-01890-f011:**
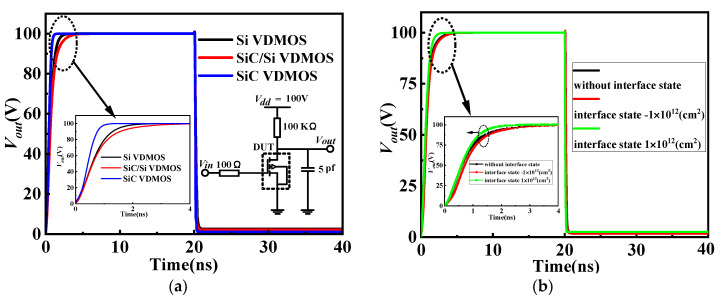
(**a**) Switching characteristics under resistive load for the SiC/Si VDMOS, Si VDMOS, and SiC VDMOS, and (**b**) the influence of different interface state charges on the switching characteristics for SiC/Si VDMOS.

**Figure 12 micromachines-14-01890-f012:**
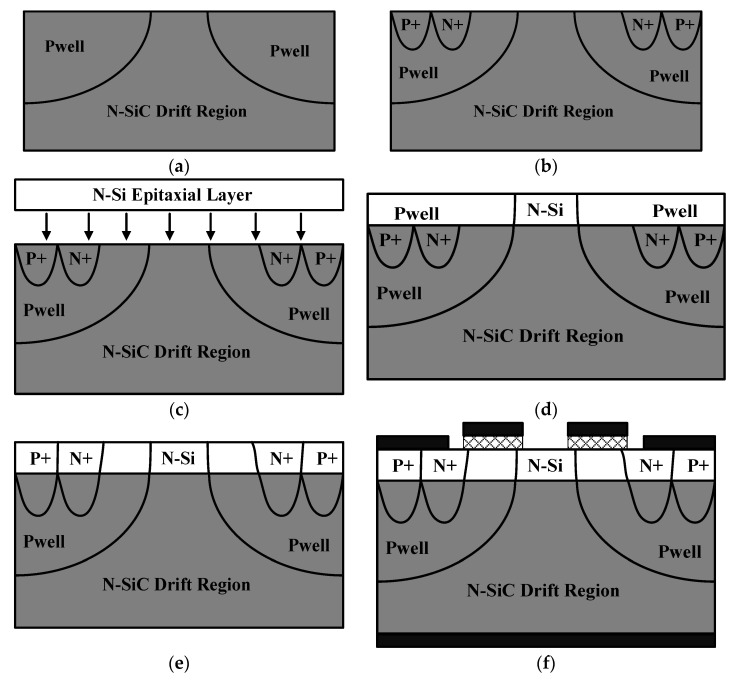
(**a**–**f**) is a simplified key process for manufacturing SiC/Si VDMOS.

**Table 1 micromachines-14-01890-t001:** Device parameters in the simulation.

Device	Si VDMOS	SiC/Si VDMOS	SiC VDMOS
*D_si_* (μm)	/	0.5	/
*T_SiC_* (μm)	/	3	/
*L_D_* (μm)	12	12	12
*N_D_* (cm^−3^)	6 × 10^14^	3.3 × 10^15^	5 × 10^15^
*N_p_* (cm^−3^)	5 × 10^17^	5 × 10^17^	5 × 10^17^
*N_SUB_* (cm^−3^)	1 × 10^14^	1 × 10^14^	1 × 10^14^

*L_D_* is the length of N-drift region; *T_SiC_* is the depth of the SiC at the P-base region; *N_D_* is the concentration of N-drift region; *N_SUB_* is the concentration of P-substrate; *N_P_* is the concentration of P-base region.

**Table 2 micromachines-14-01890-t002:** Simulation results for the Si VDMOS, SiC/Si VDMOS, SiC VDMOS, and GaN/Si VDMOS.

Device	Si VDMOS (*L_D_* = 12 μm)	SiC/Si VDMOS (*L_D_* = 12 μm)	SiC VDMOS (*L_D_* = 12 μm)	GaN/Si VDMOS (*L_D_* = 20 μm)
*BV*(V)	259	1144	1747	2029
*R_on,sp_* (mΩ·cm^2^)	18.2	6.03	5.03	17.2
*FOM* (MW/cm^2^)	3.68	>217	607	2939.4

## Data Availability

Not applicable.

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
