# Peer review of "Analysis of SiC/Si Heterojunction Band Energy and Interface State Characteristics for SiC/Si VDMOS"

_micromachines, 2023, doi:10.3390/mi14101890_

Round 1
Reviewer 1 Report
This article proposes the use of SiC/Si heterojunction to optimize the parameters of VDMOS, which increases the breakdown voltage while reducing the on-resistance. At the same time, this article conducts a comprehensive analysis of the heterojunction interface state characteristics, allowing readers to have a deeper understanding of optimized VDMOS. But there are still something that need to be improved.
1. Please clarify the definition of the device parameter TGaN in Table 1. Why use parameter of GaN, since the topic of this manuscript is SiC?
2. Please unify the description of Figure or Fig. in the whole text.
3. The interface description of the four lines AA'-DD' is very vague. According to the inset in Fig. 5, DD' appears to be the N--Si/N--SiC interface, please explain this.
4. The temperature of the device is different under different working conditions. Please clarify under which conditions the simulation results are obtained in Fig. 7.
5. There should be a space between numbers and units in the text. Please check the writing standards for the full text including figures.
Minor editing of English language required.
Reviewer 2 Report
This paper investigated the influences of interface state on the performances of the SiC/Si VDMOS. But it should be revised before publishing. The following comments might help in improving the technical quality of the manuscript.
1. The SiC/Si VDMOS is proposed in your previous paper. This paper discussed the influences of interface state on the electric field, band energy distributions and so on. Therefore, the sentence of “a SiC/Si VDMOS is proposed in this paper” in the abstract and conclusions should not be used.
2. The “Heterojunction Band” in the title is never mentioned in the main text. The title should stick to the main content.
3. The comparison is SiC/Si VDMOS to the Si VDMOS, it is more reasonable if authors give the SiC VDMOS to make the comparison.
4. The TGaN in Table I should be given its meaning and marked in Fig.1.
5. In line 109-111, this sentence is obtained from Fig. 3(a), but not from the references [16-24], authors should make sure the citation is correct.
6. The titles of section C and section D are same. Please check this mistake.
7. Fig.12 only give the turn-off characteristic, it is better to give the turn-on characteristic.
8. The name of Si VDMOS and Con.Si VDMOS should be consistent.
9. The font of the coordinate axis in the figures are too large.
Reviewer 3 Report
The paper assumes the feasibility of a complicated device. For this reason the submitted publication is very speculative, because it is assuming ideal Si channel material. In reality this Si will convert, due to high temperature processing into a defected Si. This in turn increases leakage currents and reduces channel mobility and the critical field strength. Another drawback of the device schematics is the fact that the p-dopant in Si is boron and in SiC it is aluminium. Therefore, to achieve this junction profile will be very difficult if not impossible. Another point to mark is that standard annealing temperatures for implant activation in SiC are above the melting point of Si. This has to be mentioned in the paper and solution have to be speculated.
The authors have not only to argue that this device structure can overcome purely Si based devices, but they have to speculate have this devices advantages compared to pure SiC devices justifying this more complex technology.
Why in Fig. 2 and Fig. 8a discontinuities in the electric field distributions are missing?
Figure 5 is overloaded. Please redesign.
The used and not given in the manuscript traps and their characteristics have to be given and motivated by experimental evidences
All assumptions and simplifications have to be given clearly have to be given to guide the reader and deepen the understanding what was done
Band structure in Fig. 5, Fig. 6 and Fig. 9 is questionable, because the discretisation step was chosen to large. This cause a non-acceptable picture of the band diagrams.
Fig. 2, Fig. 8a were the drop of the electric field slightly above 8 µm comes from? Why he appears at 15.6 µm in Fig. 2b.
Rounding of the obtained numbers have to be done to a reasonable value. Four digits are useless. Some examples are given in the pdf-file.
Please check for typing errors. Some are highlighted in the text.
Please improve the text. First describe what is given in the Figs and then discuss the physical reason, at the end make conclusions
Round 2
Reviewer 2 Report
Authors have revised the manuscript according to the comments. But for the comparison, authors only give the BV, Ron,sp and FOM of the Si VDMOS, SiC/Si VDMOS, SiC VDMOS and Gan/Si VDMOS. It seems that the performance of SiC/Si VDMOS is much lower than the SiC VDMOS. This manuscript is focus on the the influence of interface state on electric field, band energy distribution, temperature characteristics, and switching characteristics. Does these characteristics better than the SiC VDMOS? It is more appropriate if the SiC VDMOS is considered in Fig.2(a), Fig.4, Fig. 7, Fig.10(a) and Fig.12(a). In additional, the Table 1 and Table 3 can be merged to one table.
Reviewer 3 Report
No comments anymore
Author Response
No comments anymore